# High-resolution two-dimensional electronic spectroscopy reveals the homogeneous line profile of chromophores solvated in nanoclusters

Ulrich Bangert [1], Frank Stienkemeier [1] & Lukas Bruder [1✉]

Doped clusters in the gas phase provide nanoconfined model systems for the study of system-bath interactions. To gain insight into interaction mechanisms between chromophores and their environment, the ensemble inhomogeneity has to be lifted and the homogeneous line profile must be accessed. However, such measurements are very challenging at the low particle densities and low signal levels in cluster beam experiments. Here, we dope cryogenic rare-gas clusters with phthalocyanine molecules and apply action-detected two-dimensional electronic spectroscopy to gain insight into the local molecule-cluster environment for solid and superfluid cluster species. The high-resolution homogeneous linewidth analysis provides a benchmark for the theoretical modelling of binding configurations and shows a promising route for high-resolution molecular two-dimensional spectroscopy.

[1] Institute of Physics, University of Freiburg, Hermann-Herder-Str. 3, 79104 Freiburg, Germany. ✉email: lukas.bruder@physik.uni-freiburg.de

The control and spectroscopic study of nanoconfined systems is of high relevance in many fields of research, ranging from quantum technologies to light harvesting applications, catalysis and spectroscopy[1–4]. To understand the structure–function relationship on the nanoscale, individual structural configurations have to be resolved with high resolution, while, ideally, parasitic environmental perturbations are minimized. These challenges are best met in studies of isolated nano-objects in the gas phase.

In cluster-isolation spectroscopy rare-gas clusters serve as nanoconfined matrix to isolate individual spectroscopic probes in the gas phase[5–9]. Helium (He) clusters form superfluid nanodroplets which efficiently cool dopant species to sub-Kelvin internal temperatures inside a homogeneous solvent, providing ideal conditions for high-resolution spectroscopy[10,11], e.g. of isolated molecules[5], radicals[7], aggregates[12], and complexes[13–15]. More recently, solid rare-gas clusters were employed as a nanoconfined cryogenic substrate facilitating the formation of molecular networks with tunable interaction strength on the cluster surface[16]. This approach has revealed cooperative molecular mechanisms such as superradiance and singlet fission with spectral resolution not achievable in thin film and bulk experiments[16,17]. While the cluster-isolation technique was so far predominantly used for high-resolution spectroscopy of embedded species, the heterogeneous cluster systems may also serve as models for the study of structural configurations in isolated nanoconfined systems.

Insight into system–bath interactions, e.g. the coupling to the bath modes, the co-existence and configuration of different binding sites and their dynamic rearrangement are generally gained from the absorption line profiles[18]. Thereby, the key challenge is the extraction of the homogeneous line profile from the inhomogeneously broadened ensemble response. However, most methods for homogeneous linewidth retrieval[18] are not compatible with the low particle densities of gas-phase cluster samples. Hole burning, as an exception, has been applied to doped He nanodroplets, however did not resolve the homogeneous linewidth[19]. This technique faces the difficulty of adapting the laser parameters to the time and frequency scales of the target system, reducing the approach mainly to photochemical hole burning where photoreactive chromophores are probed on quasi-infinite time scales[18]. These challenges may explain why in heterogeneous cluster samples the homogeneous linewidth has not been determined, so far.

Recently, two-dimensional electronic spectroscopy (2DES) has been established, which is an ultrafast nonlinear spectroscopy technique enabling the disentanglement of homogeneous and inhomogeneous broadenings while automatically adapting to the time–frequency scale of the probed ensemble[20]. 2DES and 2D infrared spectroscopy[21] have proven as very useful in condensed-phase systems to extract line shape information where other methods are not applicable or do not provide the required time–frequency resolution[22–26]. Due to technical challenges, the method's potential in gas phase experiments is hardly explored[27–29].

Here, we apply the method to a cluster-isolated chromophore in the gas phase, which enables us to resolve the homogeneous absorption profile of the system. As chromophore we chose free-base phthalocyanine ($H_2Pc$), which belongs to a class of aromatic molecules of high relevance in optoelectronics, nonlinear opical materials and photobiology[30–34]. Our results provide insight into the molecule-surface binding configurations with a resolution far beyond the accuracy and resolution of current density functional theory approaches. This offers a perspective for resolving the role of local configurations in photo-chemical reactions and opens a route for ultrafast multidimensional spectroscopy studies of isolated molecular systems with high spectral resolution.

## Results

**Experimental scheme.** Figure 1 summarizes the experimental scheme. A supersonic beam of rare-gas clusters is generated in a molecular beam apparatus and is doped with $H_2Pc$ molecules (details in the "Methods" section). In our study, we compare the molecular response of a single chromophore dissolved inside a superfluid He droplet with the response of 2–3 chromophores attached to the surface of a solid Ne cluster (Fig. 1a). At the low equilibrium temperatures of the nano-systems (He droplet: 0.37 K[35], Ne cluster: ≈10 K[36]), only the lowest vibrational state of the dopant molecule is thermally occupied. Nearest-neighbor interactions between the chromophores are well-suppressed in both cluster-isolation experiments and no spectral signatures (line shifts/splittings) of inter-molecular couplings are observed despite the high spectral resolution of the experiment.

So far, 2DES is mainly performed in the condensed phase and the desired nonlinear signals are separated from the background by non-collinear four-wave mixing geometries (coherent-detected 2DES)[37]. Conversely, for the dilute cluster beam samples a collinear beam geometry is needed combined with the detection of an action signal (action-detected 2DES)[29]. In the latter, the sample is excited with a sequence of four femtosecond laser pulses (Fig. 1b) and the fourth-order light–matter response is deduced from the detected fluorescence. The pulse delays $\tau$, $t$ are interferometrically scanned to track the free polarization decay of the sample induced by pulses 1 and 3, respectively. Accordingly, a Fourier transform with respect to $\tau$, $t$ yields 2D frequency-spectra, which directly correlate the excitation ($x$-axis) and detection ($y$-axis) frequencies, while the time delay $T$ determines the time evolution of the correlation spectra (Fig. 1c, d)[20].

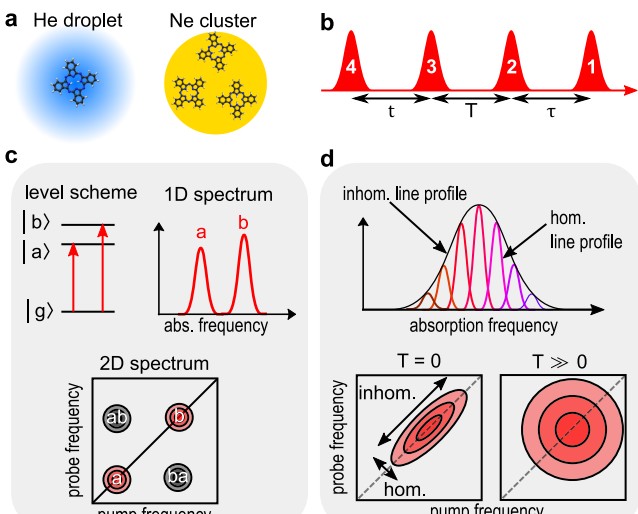

**Fig. 1 Experimental scheme. a** Studied sample types: single $H_2Pc$ molecule dissolved in a He droplet (blue) and ≤3 molecules attached to the surface of a solid Ne cluster (yellow). **b** Optical femtosecond pulse train used in the 2DES experiment. **c, d** Relevant signal contributions in the 2D spectra. In **c** the linear excitation spectrum is mapped onto the diagonal of the 2D spectrum (peaks labeled a, b), while nonlinear couplings between the quantum states (e.g. through a common ground state) lead to off-diagonal peaks (labels ab, ba). In **d** depending on the local ensemble configuration, the homogeneous absorption of the molecules is frequency-shifted, leading to an inhomogeneously broadened lineshape. In the 2D spectra, the inhomogeneous (homogeneous) lineprofile is aligned along the diagonal (anti-diagonal), and their correlation can be studied as a function of the evolution time $T$.

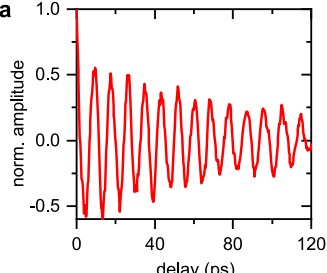
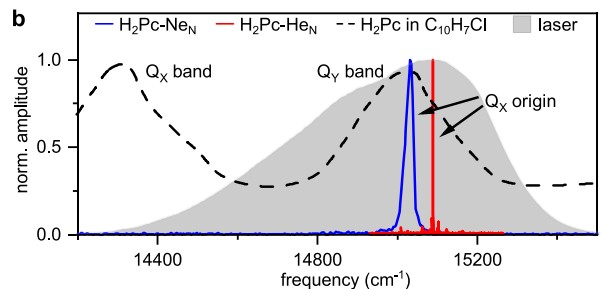
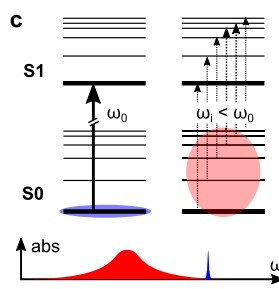

**Fig. 2 1D coherence scan of cluster-isolated H₂Pc. a** Long-lived electronic coherence of $H_2Pc$–$He_N$. The coherence oscillation is frequency-shifted to a lower domain due to rotating frame detection. In return, the frequency axis in the Fourier spectra in **b** are up-shifted by the same amount to recover the absolute transition frequencies of the system (see the "Methods" section). **b** Steady-state absorption spectrum of $H_2Pc$ solvated in 1-chloronaphthalene at room temperature (22 ± 1 °C)[75] (dashed black) compared to the Fourier spectra of the 1D coherence scans for the $H_2Pc$–$He_N$ (red) and $H_2Pc$–$Ne_N$ (blue), used femtosecond laser spectrum (gray). The shown $H_2Pc$–$He_N$ linewidth is limited by the experimental resolution (see the "Methods" section). **c** Excitation scheme of cold (blue) and hot (red) $H_2Pc$. The shaded area illustrates the thermal population of vibrational states. For hot molecules many vibronic absorption lines exist leading to a broadened red shifted absorption spectrum relative to the one obtained for cold molecules, as schematically shown in the absorption spectrum at the bottom.

The ultralow optical density of doped cluster beam samples[27], requires a highly sensitive 2DES apparatus[29], which adapts the phase modulation technique developed by Marcus and co-workers[38]. This method implements carrier-envelope-phase modulation of the optical pulses on a shot-to-shot basis at a high laser repetition rate (200 kHz). The phase modulation leads to characteristic beat notes in the fluorescence yield based on which the linear and nonlinear signal contributions are efficiently separated and amplified using lock-in detection. Thereby, the lock-in detection suppresses phase noise in the interferometric measurement and greatly enhances the detection sensitivity (details in the "Methods" section). These properties enabled several studies of highly dilute samples[27,39,40] and quantum interference measurements at extremely short wavelengths[41,42].

The relevant information content of the 2D spectra is schematically shown in Fig. 1c, d. Peaks on the diagonal position of the 2D spectra closely resemble the linear excitation spectrum of the system, whereas off-diagonal peaks directly disclose nonlinear couplings, which are normally obscured in linear absorption measurements (Fig. 1c)[43]. In addition, 2D lineshapes provide insight into ensemble inhomogeneities and their dynamic evolution (Fig. 1d). With the coherent multipulse excitation scheme, nonlinear rephasing and non-rephasing signals are recorded which leads to a separation of the inhomogeneous(homogeneous) lineshape along the diagonal(anti-diagonal) projection of the 2D spectra, accordingly. At time $T = 0$ fs this effect leads to an elongated peakshape indicating a strong correlation between the absorption and detection frequency. For $T > 0$ statistical fluctuations of the local environment lead to a loss of the correlation resulting in increasingly symmetric peak shapes (termed spectral diffusion). This concept is known from 2D photon echo spectroscopy as performed in coherence-detected 2DES[20,43]. The equivalence between coherence and action-detected 2DES in terms of lineshape information was shown in ref. [44]. We note that coherence and action-detected 2DES may differ in the detection of excited state absorption and multiple quantum coherence signals[45,46] which are, however, not relevant in the current study.

**1D coherence scans.** So far, the majority of 2DES studies have been performed in the liquid and solid phases[29]. To relate our approach to experiments in the condensed phase, we perform 1D coherence scans (using pulses 1 and 2 only) of $H_2Pc$ attached to Ne clusters (denoted $H_2Pc$–$Ne_N$) and He nanodroplets (denoted $H_2Pc$–$He_N$) and compare the Fourier spectrum to the linear absorption spectrum of $H_2Pc$ in an organic solvent (Fig. 2). Figure 2a shows the 1D coherence scan of the $S_1 \leftarrow S_0$ transition in

$H_2Pc$–$He_N$, featuring a clean, long-lived oscillation of the electronic coherence in the time domain. While in the condensed phase, electronic coherences are strongly perturbed and decay typically within <100 fs[47], they persist for more than 100 ps in $H_2Pc$–$He_N$ (Fig. 2a) and ≈3 ps in $H_2Pc$–$Ne_N$ (not shown), implying a weak coupling of the cluster environments to the molecule's electronic degrees of freedom. We note, that the full coherence decay in $H_2Pc$–$He_N$ extends beyond the experimental observation window. The observation of a single oscillating frequency in Fig. 2a suggests, that the molecule is initially prepared in a single state and the laser field drives a single vibronic transition. This offers ideal conditions for coherent control applications as well as optical trapping and cooling of molecules[48,49].

A Fourier transform of the 1D coherence scans yield the linear absorption spectra of both systems (Fig. 2b). For reference we show the $H_2Pc$ absorption spectrum in a 1-chloronaphthalene solution. Comparing the molecular response in the three different environments, a clear trend can be observed. An increasing line broadening and red shift of the absorption band occurs when going from $H_2Pc$–$He_N$ to $H_2Pc$–$Ne_N$ and $H_2Pc$ in 1-chloronaphthalene. This can be explained by the reduced perturbation and lower temperature in the cluster-isolated molecular samples (cf. Fig. 2c). A zoom on the $Q_x$ absorption band of the cluster-isolated molecules is shown in Supplementary Fig. 1 along with a comparison to high-resolution steady-state laser excitation spectra.

In the $H_2Pc$–$He_N$ spectrum several sharp peaks at much lower amplitude are observable clearly separated from the $Q_x$ absorption. These lines are assigned to complexes of $H_2Pc$ and clusters of $H_2O$, $N_2$, or $O_2$ as also discussed in refs. [50,51]. While the superfluid, homogeneous He solvent favors the formation of heterostructures, complex formation on the surface of solid Ne clusters is less likely, explaining the absence of respective spectral signatures in this system.

**2D spectra.** Having compared the linear spectra to studies in solution, we turn now to the nonlinear 2DES experiments. Figure 3a shows the 2D spectrum of $H_2Pc$–$He_N$ featuring several sharp, highly resolved diagonal peaks and no discernible off-diagonal resonances. Note, that the color scale is saturated by a factor of 10 to enhance the weaker spectral features. The strong diagonal peak at 15,088.9 cm⁻¹ marks the $S_1^0 \leftarrow S_0^0$ transition in $H_2Pc$, i.e. the zero-phonon line (ZPL). The horizontal and vertical lines intersecting with the ZPL show the tails of the ZPL line profile, which are augmented by the saturated color scale. The weaker diagonal peaks are assigned to the heteromolecular

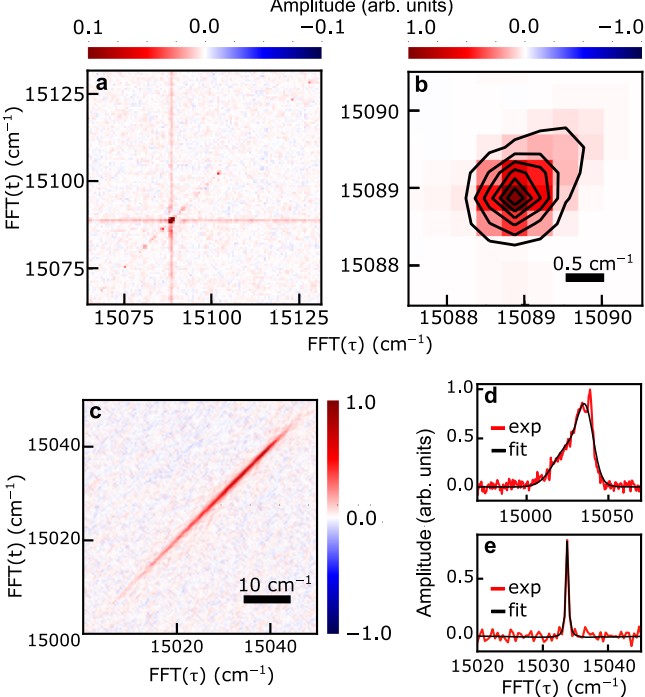

**Fig. 3 2D spectra of doped nanoclusters. a**, **b** $H_2Pc$ dissolved in superfluid helium nanodroplets and **c** $H_2Pc$ adsorbed on the surface of solid neon clusters (all at $T = 1\,ps$). In **a** the color scale is saturated by a factor of 10, **b** shows an unsaturated zoom of the same data with contour lines at 14.2% of the maximum amplitude. **d**, **e** show the inhomogeneous(homogeneous) line profiles obtained by a cut of **c** along the diagonal axis (for **d**) and anti-diagonal axis at 15,034 $cm^{-1}$ (for **e**). Black solid lines show the projections of the fit function obtained from a 2D lineshape fit.

complexes discussed above whose spectral positions coincide exactly with the weak features observed in the 2D spectra. Possible contributions from sample scattering can be excluded in action-detected 2DES[52]. The absence of cross-peaks confirms that these spectral features correspond to individual molecular species which do not couple among each other (cf. Fig. 1c). Since in the cryogenic, superfluid matrix clustering of dopant molecules is unavoidable, the absence of cross-peaks imply that the molecular complexes each occupy a separate nanodroplet. Figure 3b shows a zoom on the 2D spectrum (unsaturated color scale), revealing the ZPL in more detail. The excitation and detection-frequency resolution in this 2D spectrum is 0.65 $cm^{-1}$ (for resolution limit see "Methods" section) which is, to the best of our knowledge, much higher than in any previously reported molecular 2DES study. Even higher resolution may be achieved with frequency comb-based 2DES[53]. However, so far, this approach has been only demonstrated for thermal atomic vapors and not for cold molecules. A spectral shoulder extending to the high-frequency side is observable which reveals the contribution of a $C_{13}$-isotope of $H_2Pc$ located at 15,089.5 $cm^{-1}$ (see Supplementary Note 1). Taking this contribution into account, the ZPL reveals a homogeneous line profile (spherically symmetric 2D peak shape) within the experimental resolution. This confirms the high homogeneity of the quantum fluid environment also predicted by recent density-functional theory calculations[54].

As already deduced from the 1D absorption spectrum, the 2DES data confirms that the system behaves in good approximation like a two-level system with only a single optical transition and no coupling to other states (absence of cross-peaks). This is remarkable for organic molecules and is a consequence of the narrow Franck–Condon window for the

$S_1 \leftarrow S_0$ transition in combination with the low internal temperature of the molecule.

For $H_2Pc–Ne_N$ the 1D absorption spectrum (Fig. 2b) reveals a much broader line profile reminiscent of systems with multiple vibronic transitions. Figure 3c shows the respective 2D spectrum of $H_2Pc–Ne_N$. Here, we observe a single, strongly elongated diagonal peak. At the cryogenic temperatures of the Ne cluster environment, only the lowest vibrational state of $H_2Pc$ is populated. The absence of cross-peaks thus implies the coupling of the ground state to a single vibronic transition in analogy to the $H_2Pc–He_N$.

The broad linear absorption spectrum can be thus purely attributed to ensemble inhomogeneities. A quantitative 2D lineshape analysis yields the inhomogeneous and homogeneous line profiles of the $S_1 \leftarrow S_0$ transition in $H_2Pc–Ne_N$. To this end, we performed a 2D peak fit by adapting the model from ref. [55] (details in Supplementary Note 2). The 1D projections of the fit result are shown in Fig. 3d, e. We find excellent agreement between the fit model and the experimental data, except for an outlier at 15,040 $cm^{-1}$. This outlier is not observed in our other data (Fig. 2) and is attributed to an experimental artifact.

We deduce for the inhomogeneous broadening a value of 23 $cm^{-1}$ and a remarkably narrow homogeneous broadening of 0.42 ± 0.01 $cm^{-1}$ (both full-width at half maximum values). These values are not limited by the resolution of the experimental apparatus. To the best of our knowledge, this is the first experimental determination of the homogeneous broadening in such hetero nanosystems. We experimentally determined the fluorescence lifetime to be > 10 ns (details in the "Methods" section), in good agreement with fluorescence lifetimes in bulk rare gas matrices[56]. This implies a negligible lifetime contribution to the homogeneous linewidth. The homogeneous broadening thus reflects in good approximation the pure dephasing rate of the system[57], which, in analogy to studies in bulk matrices, is attributed to an elastic scattering with phonon modes of the rare gas cluster[18,57]. A sample-temperature-dependent study of the homogeneous linewidth would provide further insight into the phonon scattering as commonly done in the condensed phase[18]. However, the infrared inactivity and rapid evaporative cooling of the nanoclusters prevent any means for heat injection.

The high-resolution data separating the homogeneous and inhomogeneous linewidth allows us to estimate a lower limit for the number of binding configurations between the $H_2Pc$ molecule and the cluster surface. To this end, we fitted the inhomogeneous lineshape with a simplistic model assuming an equidistant spread of homogeneous line profiles. Fit parameters were the density of homogeneous absorption lines $n_L$ and their amplitudes $A_i$. We optimized the fit by gradually increasing the line density and considered as convergence criterion the point where the spectral modulations in the fit function are on the level of the noise of the experimental data (Fig. 4). We find that a minimum line density of $n_L \geq 4.0$ lines/$cm^{-1}$ (corresponds to 360 configurations) is necessary to fit the data. Taking the finite spectral resolution of the experiment into account, this constrains the number of binding configurations to a lower limit of ≥216 (details in Supplementary Note 3), which corresponds to a mean energetic separation of the binding configurations of only 0.42 $cm^{-1}$.

This information about binding configurations is clearly beyond the resolution and accuracy of current theory and may help to gauge new models of molecule-surface binding configurations, as they play an important role in, e.g. surface chemistry. Here, we attribute the large number of binding configurations to the different geometric orientations of the molecule with the cluster lattice and to cluster surface defects (icosahedral cluster structure[36]), while only a minor effect is expected from the statistical cluster size variation in the probed ensemble. The latter is rationalized by the fact, that the current study employed fairly

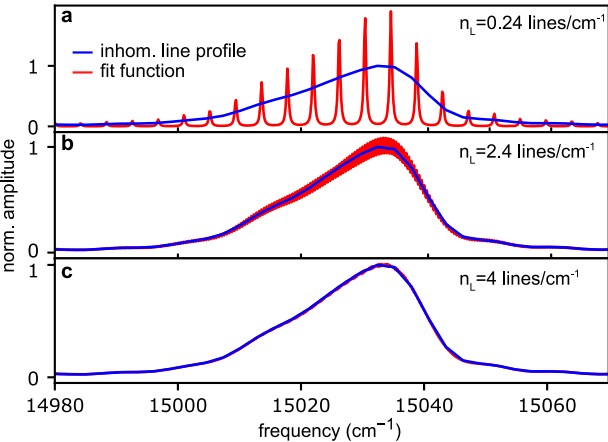

**Fig. 4 Fit of molecule-surface binding configurations.** Experimental inhomogeneous line profile (blue) and simplistic fit model of equidistantly spaced homogeneous line profiles (red). By increasing the line density $n_L$ from **a** to **c**, the fit converges. The 1D coherence scan was used for the experimental data, featuring a lower noise level then the 2DES data.

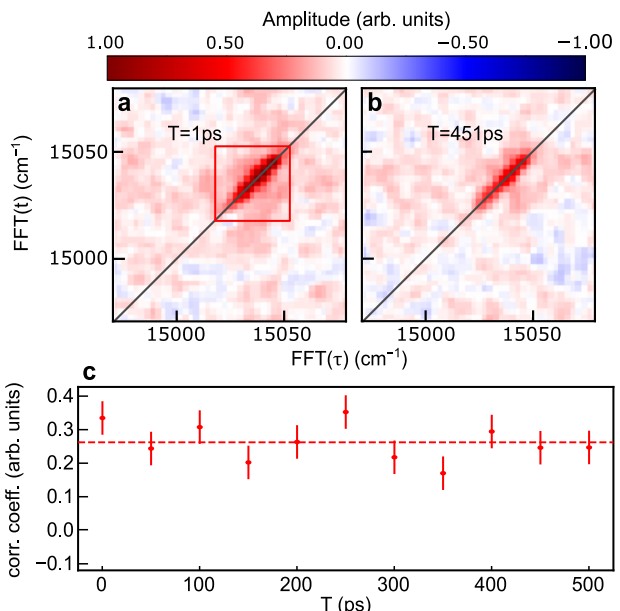

**Fig. 5 Time evolution of $H_2Pc–Ne_N$. a, b** 2D spectra at evolution times $T = 1$ ps and 451 ps, recorded at reduced spectral resolution. **c** Correlation coefficient calculated for the marked area in **a** as a function of the evolution time $T$. The dashed line shows the mean correlation value.

small clusters ($\approx 400$ Ne atoms per cluster[58]), while basically the same inhomogeneous line profile was obtained for much larger cluster sizes (see nanosecond laser excitation experiment in Supplementary Fig. 1). In contrast, experiments in bulk Ne matrices showed an almost one order of magnitude broader inhomogeneous linewidth of $\approx 200$ cm$^{-1}$[56]. The same behavior is observed for 3, 4, 9, 10-perylenetetracarboxylic dianhydride (PTCDA)[59]. The much broader inhomogeneous linewidth in the bulk environment may be explained by the formation of a solvation shell when the molecule is fully embedded in the bulk matrix compared to when bound to the surface of a cluster. At the same time, the homogeneous broadening of $H_2Pc–Ne_N$ seems in reasonable agreement with the values obtained in bulk matrices of heavier rare-gases where a factor of $\approx 5$ smaller homogeneous broadening is observed[60]. The reduced phonon broadening may be explained by the lower matrix temperature (4 K) as compared to the cluster samples with temperatures of $\approx 10$ K.

2DES also offers time-resolved information about binding configurations (cf. Fig. 1d). The principle is well established in the condensed phase, where thermal fluctuations in the surrounding bath lead to a rearrangement of the binding configurations over time[23,61]. The spectral diffusion of 2D lineshapes thus mainly reflects the solvent dynamics. In contrast, in the cryogenic cluster environment thermal fluctuations are absent and changes in the binding configurations may be directly attributed to the dynamics of the dopant molecules. In this context the question arises, if an electronic excitation of the dopant can change the binding configuration with the environment. Such mechanisms are common for species dissolved in He droplets, leading, e.g. to a migration towards the droplet surface followed by the ejection of the dopant on a pico to femtosecond time scale[62,63].

In the 2D spectra of $H_2Pc–Ne_N$, the onset of any configuration change would be observable by a broadening of the anti-diagonal linewidth or a reduction of the correlation between excitation and detection frequencies with increasing time $T$. We studied the time evolution of the hetero-nanosystem for an evolution time of 0–500 ps at a reduced spectral resolution (2.3 cm$^{-1}$ for anti-diagonal linewidth) to limit the measurement time (Fig. 5). No significant temporal changes can be observed (Fig. 5c), implying that fluctuations in the binding configurations are below the experimental resolution of 2.3 cm$^{-1}$ (0.3 meV). This accounts for molecules populated in the electronic ground and excited state, since the response of both is reflected in the 2D spectra. We

conclude that the molecules are mainly immobile on the cluster surface on the time scale of 500 ps, which was previously only indirectly confirmed for ground state molecules[59]. We note, that the rotation of the cluster in the laboratory frame adds in principle another contribution to the inhomogeneous broadening due to the cluster size distribution in the probed ensemble. This broadening is however expected to be negligible due to the large moment of inertia of the clusters.

On the contrary, dopant species are highly mobile on the surface of superfluid He droplets as reflected in the formation of alkali complexes on the droplet surface[64]. Likewise, species fully immersed in the quantum fluid clusters exhibit high mobility. Their rotational motion is well studied and served as a probe of the internal temperature of He clusters[35]. The solvation mechanisms of organic chromophores in superfluid He droplets have been previously studied with high-resolution frequency-domain spectroscopy[65]. It was rationalized that solutes are embedded in a non-superfluid He solvation layer. The different configuration of the solvation complexes lead to spectral finestructures[19,66] while the dispersive solute–cluster interaction causes asymmetric lineshapes due to the statistical cluster size distribution[50]. The homogeneous linewidth for the ZPL of solutes in superfluid He has not been determined to date. High-resolution 2DES may solve this problem and provide new insight into the solvation mechanisms in superfluid media. For Pc-He$_N$ an inhomogeneous linewidth of $\geq 0.1$ cm$^{-1}$ was found and an estimate for the homogeneous broadening of 0.02 cm$^{-1}$ was indirectly deduced[50]. Both are beyond our current experimental resolution. This could be solved by deploying advanced sampling strategies[67–69] or implementing frequency-comb technology[53] in order to increase the resolution to the required regime, thus, opening a promising perspective for the study of solvation mechanisms in superfluid environments.

In conclusion, the current study explored the nonlinear response of isolated chromophores in different nanosystems with femtosecond time resolution. The unique combination of 2DES with cluster-isolation techniques uncovered the homogeneous lineprofile beyond the ensemble average, providing insight into

the molecule-surface binding configurations with high resolution. Other methods for homogeneous linewidth retrieval cannot provide the combined time–frequency resolution of 2DES. The presented high-resolution 2DES approach, thus, provides a promising perspective for the analysis of molecular dynamics as a function of the local environment with femtosecond time resolution. Conversely, the here studied nanosystems represent highly resolved, well-defined molecular two-level systems which offer ideal conditions to explore many-body mechanisms of high interest in photovoltaics, such as molecular exciton migration and annihilation processes, as well as in quantum information science which is, so far, predominately limited to atomic samples. The potential of studying molecular networks on nanoconfined cluster surfaces was recently demonstrated[16,17] and the advantage of 2DES in uncovering inter-particle interactions and cooperative dynamics was proven in previous studies[70–72].

## Methods

**Sample preparation**. The experimental apparatus including the cluster beam preparation are described in detail in a recent review article[29]. Further information about the sample preparation is also given in refs. [6,73]. Briefly, a continuous jet of the rare gas is expanded into vacuum with a stagnation pressure of 50 bar through a cooled nozzle (5 μm orifice). The nozzle temperature is 14 K for He (mean cluster size of $N_{He} = 15,000$ atoms[5]) and 70 K for Ne ($N_{Ne} = 400$[58]), respectively. The cluster beam is skimmed by a 400 μm conical skimmer positioned in a distance of 15 mm from the nozzle. The cluster beam is doped in the adjacent chamber, where it passes a 10 mm-long, heated oven cell containing $H_2Pc$ powder (Sigma Aldrich, 29H,31H-Phthalocyanine, 98%). The doping follows a statistical pick-up process described by Poissonian statistics and is controlled by the cell temperature and, thus, the density of evaporated molecules. The cell temperature is optimized for maximum fluorescence signal, which corresponds to molecule densities of $5 \times 10^{12}$ cm$^{-3}$ (350 °C cell temperature) for He and $3.2 \times 10^{13}$ cm$^{-3}$ (380 °C) for Ne, respectively. At these conditions, the estimated mean dopant number per cluster are 1 for He and 2–3 for Ne clusters. Evaporation of cluster atoms cools the nanosystems afterwards to equilibrium temperatures of 0.37 K for He[35], and ≈ 10 K for Ne[36]. The base pressure in the doping chamber was $1.48 \times 10^{-6}$ mbar which is due to the effusively emitted hot molecules from the oven cell. The doped cluster beam passes an orifice of 5 mm into the next chamber to suppress the latter contribution. In this chamber ($3.8 \times 10^{-8}$ mbar base pressure) the femtosecond laser pulses intersect with the cluster beam at right angle and the sample fluorescence is imaged with a lense onto a photo multiplier tube mounted in perpendicular direction to the laser and cluster beam propagation. The detector arrangement collects ≈17% of the total fluorescence. The detector response time is 0.57 ns. The fluorescence lifetime was determined by tracking the decay of the fluorescence yield with a fast digital-to-analog converter (bandwidth: 500 MHz, sampling rate 1 Gs/s).

**Spectroscopy method**. The high-repetition-rate femtosecond laser system and phase-modulated 2DES setup is described in detail elsewhere[29,74]. Laser parameters were: center wavelength 660–670 nm, spectral width 25 nm (FWHM), pulse duration ≈50 fs, pulse energies 25–35 nJ per pulse, laser focus diameter 200 μm (both at the target) and laser repetition rate 200 kHz. A typical pulse spectrum is shown in Fig. 2b.

For the 2DES experiments, the cluster beam was excited by a collinear pulse train of 4 fs laser pulses and the fluorescence is detected as parametric function of the inter-pulse delays $\tau$, $T$, $t$. A 2D Fourier transform with respect to $\tau$, $t$ yields the rephasing and non-rephasing 2D spectra of which the absorptive correlation spectrum is computed and shown in the main text, except for Fig. 3c showing the real-part of the rephasing spectrum. For the 1D coherence scans pulses 3,4 were blocked and the real-part of the 1D Fourier transform with respect to $\tau$ is shown in the main text. For the 1D coherence scan in Fig. 2, $\tau$ was scanned from 0 to 120 ps in 100 fs steps. For the 2DES measurements the coherence times $\tau$, $t$ were scanned, from 0 to 50 ps in steps of 500 fs (Fig. 3a, b), from 0 to 42 ps in steps of 300 fs (Fig. 3c) and from 0 to 6 ps in steps of 300 fs (Fig. 5). For the measurements in Fig. 5 the population time $T$ was scanned from 1 to 501 ps. Values $T < 100$ fs were omitted to avoid the influence of parasitic pulse overlap effects.

To increase the signal-to-noise ratio, the carrier-envelope phase of each pulse is modulated at frequencies $\Omega_i (i = 1–4) \approx 110$ MHz on a shot-to-shot level using acousto-optical modulators. This leads to well-defined beat notes in the fluorescence signals of $\Omega_{21} = 5$ kHz, $\Omega_{43} = 8$ kHz for the linear excitation (1D coherence scan) and $\Omega_{21} \pm \Omega_{43} = 13$ kHz for the fourth-order non-rephasing and 3 kHz for the rephasing signal, respectively ($\Omega_{ij}$ denotes $\Omega_i - \Omega_j$). The beat signals are efficiently filtered and amplified by lock-in detection.

As reference signal for the lock-in detection, a portion of the pulse train is spectrally filtered in a monochromator (0.025 nm spectral resolution) at a frequency of $\nu_R = c/662.33$ nm and detected with a PMT. The reference signal records the optical interference of the pulses and, thus, tracks the phase changes and phase jitter in the optical setup at the frequency $\nu_R$. The heterodyne

demodulation of the fluorescence signal $S(\tau, T, t, \Omega_{21} \pm \Omega_{43})$ with the reference signal $R(\tau, T, t, \Omega_{21} \pm \Omega_{43})$ inside the lock-in amplifier removes the modulation as well as the phase jitter from the signal $S$, which provides the required interferometric stability for the measurements. In addition, $S$ is detected in the rotating frame of $R$, which down-shifts the signal frequencies $\nu_S$ by 2 orders of magnitude to $\bar{\nu} = \nu_S - \nu_R$. To recover the absolute frequencies in the Fourier spectra, the frequency axis are up-shifted again by $\nu_R$.

For the given monochromator resolution, the reference interference signal is detectable for $\tau$, $t$ delays up to 120 ps, which determines the attainable Fourier resolution in the experiment. To avoid Fourier-transform artifacts, the 1D coherence scan data of the $H_2Pc$–He$_N$ sample was apodised in the time domain with an exponential function decaying to 5% of its amplitude at the end of the delay scan range. The 2DES data of the $H_2Pc$–He$_N$ sample was apodised by a 2D Gaussian decaying to 5% of its amplitude along each dimension. The apodisation leads to an artificial broadening of the spectral lines reflecting the limited resolution of the experimental apparatus. Accordingly, for the 1D coherence scan, the experimental resolution was 0.3 cm$^{-1}$ and for the 2DES measurements (reduced scan range) it was 0.65 cm$^{-1}$ (Fig. 4) and 4.6 cm$^{-1}$ (Fig. 5) in each Fourier dimension. Note, that the resolution along the diagonal (anti-diagonal) dimension is higher[55]. In case of the $H_2Pc$–Ne$_N$ sample, data apodisation was not necessary and the inhomogeneous (homogeneous) linewidths of the sample were determined without spectral broadening introduced by the experimental apparatus.

## Data availability

The data that support the findings of this study are available from the corresponding author upon request.

## Code availability

Code used to process the data of this study is available from the corresponding author upon request.

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

## Acknowledgements

Funding by the European Research Council within the Advanced Grant "COCONIS" (694965) and by the Deutsche Forschungsgemeinschaft (IRTG 2079) are acknowledged. The article processing charge was funded by the University of Freiburg in the funding program Open Access Publishing. We thank M. Walter and S. Ferchane for useful discussions about DFT calculations of the Ne–$H_2$Pc interaction and we thank A. Slenczka for supplying the raw data of the reference spectrum shown in Supplementary Fig. 1.

## Author contributions

L.B. and F.S. conceived the experiment. U.B. preformed the measurements. U.B. and L.B. analyzed the data. L.B. wrote the manuscript with input from all other authors.

## Funding

## Competing interests
The authors declare no competing interests.
