## [Peer Review File · Nature Communications]

Reviewers' comments:

Reviewer #1 (Remarks to the Author):

In this paper, the authors report high-resolution 2DES measurements performed on gas clusters doped with phthalocyanine molecules with the aim of determining with high precision the homogeneous line profile.

The high-resolution 2DES setup is based on previous schemes (phase modulation by Marcus) but it represents indeed a new development, especially when applied to ultracold gas nanoclusters. In this sense, the presented data are novel, interesting, and worthy to be published. However, I do not believe that the quality of the analysis and the impact of the results are suitable for Nature Comm. Moreover, I have the feeling that the results and the conclusions are not yet fully mature.

Below are a few comments to support my opinion:

On page 2 the authors explain the signals measured in 2DES defining the signal as a 'photon echo'. This is wrong. They do not measure a photon echo but a fluorescence. Moreover, they discuss the data (also in terms of homo and inhomogeneous broadening) referring to the wide body of theories developed for 2DES in the conventional photon echo BOXCARS configuration (where the response is stimulated by 3 laser pulses and the signal is a coherent electric field, an 'echo'). Instead, what they do is an 'action-based' technique, where they employ 4 pulses. The 4th pulse brings the system into an excited state population from which fluorescence (the incoherent action signal) is recorded. The response function, in this case, is completely different from the one stimulated in the conventional photon echo experiments (see for example <https://doi.org/10.1021/acs.jpcclett.9b03851>). Therefore, the analysis is based on wrong or at least inaccurate foundations. (an additional minor note in this context: also figure 1b is misleading because they define time intervals like in BOXCARS photon echo 2DES).

About the impact of the results. I feel like the real exciting outcome of the paper is the possibility of pushing the resolution of multidimensional techniques. I do not find particularly impactful the determination of homogenous linewidth, which could have been measured with much easier techniques.

Two of the advantages of using the complex 2DES technique rather than simpler monodimensional techniques are: (i) the possibility of achieving frequency and time resolution and (ii) getting info on the couplings through the appearance and the time dependence of cross-peaks. None of these

capabilities was exploited here. No cross-peak appeared in the maps and basically no dynamics was recorded (the time evolution of the studied systems didn't really show interesting features, which was expected considering the experimental conditions). Basically, a complex multidimensional technique has been downgraded to a static technique, and only two maps (figure 3) out of multidimensional matrixes (which I guess required a long time to be acquired) were effectively used to draw the conclusions.

In addition, the title and the introduction claimed an investigation on the role of the environment, but this is not what the paper does. There is no real comparison between different environments (except that recognizing that in Ne droplets there is a contribution by inhomogeneous broadening), nor a discussion about what the environment does on the system. Cooled gas nanodroplets are not a natural environment for molecules and therefore I found it inappropriate to claim that the role of environmental parameters affecting the dynamics could be studied in these conditions. Also because in the ultracold experimental conditions applied here there is basically no change in the system dynamics (as stated on page 5). I would suggest changing the tone of the introduction because it is bringing the reader somewhere else.

A final comment: most of the experimental details are lacking and I do not believe that enough detail is provided in the methods for the work to be reproduced.

Minor comments:

1. I suggest specifying better in the title that nanoparticles are intended as gas nanoclusters: in material science, 'nanoparticles' might indicate a completely different system.
2. I found Figures 2c and 2d a bit misleading. While I understand that the scope is illustrating in general terms the meaning of the signals appearing in the 2DES maps, one would expect to see similar features also in the experimental data. Also, I found a bit confusing the labels 'molecule 1' and 'molecule 2', considering that only one molecular species was investigated here. I suggest removing these panels, not fully relevant here (since no cross-peaks were detected), and referring to previous literature (there are many clear and comprehensive reviews about the signal distribution in 2DES maps).
3. there is a typo on page 6, second column, line 10: 'monochromoator'

Reviewer #2 (Remarks to the Author):

Overall, this is a noteworthy and pioneering study demonstrating that local environments strongly affect the optical properties of an organic molecule chromophore. The authors use their results to report on the molecular-surface binding configurations, which is a key contribution. The authors have chosen to write their manuscript from the standpoint of spectroscopy method-development, which lowers the impact somewhat. The impact would have been higher had the authors focused their title and abstract on the scientific (new knowledge) advances with less focus on the tool used to generate that new knowledge. The method development aspect for cluster-isolation and high-resolution 2D spectrometer is very strong.

Major comment:

- The authors compare measurements of helium nano droplets and neon nanodroplets. There are two differences in these systems. One is the obvious change of temperature. The second is a change of solvation configuration, fully encompassed or on the surface. It is not clear how the authors disentangle these two effects without a temperature-dependent study.
- The generality of the results could have been enhanced had the authors chosen to conduct measurements of at least one more molecule.

Minor comments:

- page 2 the authors write: "Peaks on the diagonal position of the 2D spectra reflect the linear absorption of the system, ...". This is often repeated in many publications, but it is not strictly true for all systems. There are at least two complications: (1) The input laser pulse spectrum almost always provides different weights to the peaks because it is not fully de-convoluted. (2) ESA peaks can - in some systems - appear on or near the diagonal and thereby affect the relative peak heights. The authors could reword as, for example: "Peaks on the diagonal position of the 2D spectra mostly reflect the linear absorption..."
- Fig. (1), panel (c). The caption could be enhanced by mentioning that the coupling refers to the blue dashed arrow.
- Fig. (2), panel (b). The unit of inverse centimeters is a wavenumber, not a frequency. This abuse of the words is endemic in spectroscopy, but that does not make it correct. Please adjust so that it is scientifically correct. There are at least two good options: (1) Replace the word "frequency" with "wavenumber", or (2) Adopt the approach of Tokmakoff and label as $\omega/(2\pi c)$.
- Page 3 the authors subtly introduce a control measurement of H2Pc in 1-chloronaphthalene. At what temperature were those measurements conducted?

- Fig. (3). Why did the authors choose a waiting time of 1 ps? Time zero would in principle be better because the system would have not undergone any spectral diffusion at all.

- Fig (3) panel (a). How do the authors know that the blips along the diagonal are not simply due to scatter?

- On page 4 the authors write "...except for an outlier at 15040 cm⁻¹, which is due to a nonrecurring experimental artifact." What is the meaning of this? If it only occurred once, the authors should fit and present a different dataset that does not have the artifact.

Reviewer #3 (Remarks to the Author):

Bangert et al. report photon echo study of free base phthalocyanine in helium and neon clusters. The authors found unexpectedly long electronic coherence of hundreds of picoseconds in helium and few picoseconds in neon. The photon echo technique enables attaining the homogeneous and heterogeneous linewidth. In both environments the homogeneous linewidths were found to be about 0.5 cm⁻¹, which is very narrow. I feel these results may deserve publication in NC. However, the manuscript should be revised according to the following points.

1. The authors should remove uncorroborated claims and proposals.

In abstract and in the text the authors claim unprecedented spectral resolution without giving appropriate comparisons. I am rather sure that a factor of at least ten higher resolution was achieved in early hole burning experiments in solid matrices. Similar, the resolution in a conventional cw LIF spectrum of phthalocyanine in helium shown in Fig.1 of SI is higher than obtained in the reviewed manuscript.

Throughout the manuscript the authors implied that the photon echo technique could be applied to study: "... local configurations in photo-chemical reactions and elucidating molecular dynamics in tailored nano-environments or at functional interfaces" (Introduction)

or

" the interaction of isolated molecules with metallic and semiconducting nanoclusters, of high relevance in catalysis and biomedical imaging" (Conclusions)

or

"to explore and control many-body mechanisms of high interest in photovoltaics, such as molecular exciton migration and annihilation processes, as well as in quantum information science which is, so far, predominately limited to atomic samples." (Conclusions)

There is not any discussion of the applicability of the photon echo technique to any of the above systems. Therefore, the above propositions sound speculative to the present referee.

I suggest the authors focus on the studied system and provide more physical insights into the observed effects

2. The principle of the two-dimensional electronic spectroscopy should be introduced in the main text that should be accessible for non-specialists.

3. "A Fourier transform of the 1D coherence scans yield the linear absorption spectra of both systems (Fig. 2b). (p.3)"

I feel this is not just FT, because the frequency of the oscillations in a) is much smaller than the peak frequency in b). The authors should better explain the application of the FT.

4. " Its far extending horizontal and vertical tails reflect the Lorentzian lineshape and are augmented by the saturated color scale." (p.3)

I do not see any horizontal or vertical tails in Fig. 3 (a).

5. " We conclude that the molecules are mainly immobile on the (Ne) cluster surface on the time scale of 500 ps, which was previously only indirectly confirmed for ground state molecules [57]." (p.5)

It would be nice to have some perspective on the limits imposed by the clusters and also to have measurements, such as in Fig. 5 with helium clusters. For example, although the molecules in neon are fixed in the cluster frame, the clusters nearly certainly are rotating in the laboratory frame. In case of helium clusters the molecules may rotate and translate inside owing to its superfluid nature, on the other hand the traditional phonon induced dephasing may be muted due to finite size effects.

Some more detailed discussion of the physical origin of the long coherence in clusters as compared to other previously studied condensed system will make this manuscript much stronger.

6. In methods, the modulation frequencies $\Omega(i)$ should be given. The explanation of the signal processing in the last two paragraphs is not understandable for a not specialist. It would be nice to have some equations for more clarity.

Response to reviewer comments

Manuscript NCOMMS-21-48836

Title: High-resolution two-dimensional electronic spectroscopy reveals homogeneous line profiles in isolated nanoparticles

Authors: Ulrich Bangert, Frank Stienkemeier and Lukas Bruder.

We thank all reviewers for their time and efforts in critically evaluating our manuscript and their helpful advice for improvements.

Reviewer #1 (Remarks to the Author):

In this paper, the authors report high-resolution 2DES measurements performed on gas clusters doped with phthalocyanine molecules with the aim of determining with high precision the homogeneous line profile.

The high-resolution 2DES setup is based on previous schemes (phase modulation by Marcus) but it represents indeed a new development, especially when applied to ultracold gas nanoclusters. In this sense, the presented data are novel, interesting, and worthy to be published. However, I do not believe that the quality of the analysis and the impact of the results are suitable for Nature Comm. Moreover, I have the feeling that the results and the conclusions are not yet fully mature. Below are a few comments to support my opinion:

On page 2 the authors explain the signals measured in 2DES defining the signal as a 'photon echo'. This is wrong. They do not measure a photon echo but a fluorescence. Moreover, they discuss the data (also in terms of homo and inhomogeneous broadening) referring to the wide body of theories developed for 2DES in the conventional photon echo BOXCARS configuration (where the response is stimulated by 3 laser pulses and the signal is a coherent electric field, an 'echo'). Instead, what they do is an 'action-based' technique, where they employ 4 pulses. The 4th pulse brings the system into an excited state population from which fluorescence (the incoherent action signal) is recorded. The response function, in this case, is completely different from the one stimulated in the conventional photon echo experiments (see for example <https://doi.org/10.1021/acs.jpcllett.9b03851>). Therefore, the analysis is based on wrong or at least inaccurate foundations. (an additional minor note in this context: also figure 1b is misleading because they define time intervals like in BOXCARS photon echo 2DES).

We disagree with the reviewer. The retrieval of homogeneous and inhomogeneous lineshapes is in coherent-detected and action-detected 2DES equivalent. The theoretical foundation for this is given e.g. in the seminal paper by Tan¹, which provides a clear and solid foundation for our conclusions. In contrast, the reference pointed out by reviewer#1 refers to differences in excited state absorption (ESA) contributions. However, since these signals do not contribute in our experiment and thus do not affect the interpretation and conclusions of our study, we refrained from referring to this work in our initial manuscript.

To clarify this point we have added the following statement to the manuscript (p. 2, para. 4): "This concept is known from 2D photon echo spectroscopy as performed in coherence-detected 2DES [20, 43]. The equivalence between coherence and action-detected 2DES in terms of lineshape information was shown in Ref. [44]. We note that coherence and action-detected 2DES may differ in

¹ H.-S. Tan, J. Chem. Phys. 129, 124501 (2008).

the detection of excited state absorption and multiple quantum coherence signals [45, 46] which are, however, not relevant in the current study.”

We agree with the reviewer, that on some occasions we used the term “photon echo” in an inaccurate context and have resolved this semantic inaccuracy in the revised manuscript.

About the impact of the results. I feel like the real exciting outcome of the paper is the possibility of pushing the resolution of multidimensional techniques. I do not find particularly impactful the determination of homogenous linewidth, which could have been measured with much easier techniques.

Two of the advantages of using the complex 2DES technique rather than simpler monodimensional techniques are: (i) the possibility of achieving frequency and time resolution and (ii) getting info on the couplings through the appearance and the time dependence of cross-peaks. None of these capabilities was exploited here. No cross-peak appeared in the maps and basically no dynamics was recorded (the time evolution of the studied systems didn't really show interesting features, which was expected considering the experimental conditions). Basically, a complex multidimensional technique has been downgraded to a static technique, and only two maps (figure 3) out of multidimensional matrixes (which I guess required a long time to be acquired) were effectively used to draw the conclusions.

We thank the reviewer for the appreciation of the high resolution achieved with our experimental multidimensional spectroscopy approach. However, we strongly disagree on the other points. The low particle density at which the samples have to be prepared in order to yield cold and isolated quantum systems results in ultralow optical densities ($OD \sim 10^{-11}$)² which prohibits absorption spectroscopy as well as dispersed fluorescence detection with sufficient high resolution. This prohibits the application of any of the established methods for homogeneous linewidth analysis (fluorescence line narrowing, photon echo techniques...) in our work. As such, action-detected 2DES is the only method which provides general access to the homogeneous linewidth in the studied isolated nanocluster samples. This by itself is an important achievement which exploits the unique capabilities of 2DES. Furthermore, we note that in several high impact 2DES experiments neither any off-diagonal peaks/couplings nor any dynamics were observed or even investigated³. The high impact of these studies is solely based on 2D lineshape analysis of diagonal peaks.

We have added the following statement to make this point clearer (p.1, para. 3): “Thereby, the key challenge is the extraction of the homogeneous line profile from the inhomogeneously broadened ensemble response. However, most methods for homogeneous linewidth retrieval [18] are not compatible with the low particle densities of gas-phase cluster samples. Hole burning, as an exception, has been applied to doped He nanodroplets, however did not resolve the homogeneous linewidth [19]. This technique faces the difficulty of adapting the laser parameters to the time and frequency scales of the target system, reducing the approach mainly to photochemical hole burning where photoreactive chromophores are probed on quasi infinite time scales [18]. These challenges may explain why in heterogeneous cluster samples the homogeneous linewidth has not been determined, so far.”

The reviewer also complained that “the time evolution of the studied systems didn't really show interesting features, which was expected considering the experimental conditions”. Recent time-resolved experiments in He droplets have indeed shown a drastic change in the solute-solvent configuration upon electronic excitation leading to ultrafast dynamics. As such, the outcome of our

² L. Bruder et al., Nat Commun 9, 4823 (2018).

³ Science 333, 1723 (2011), Nat Photon 9, 663 (2015), Nat Commun 9, 2519 (2018), ACS Nano 15, 6499 (2021), Sci. Adv. 7, eabf4741 (2021).

time-resolved study was not clear and adds in our opinion valuable information to the manuscript. We made this clearer in the revised manuscript (p. 5, para. 3): “In this context the question arises, if an electronic excitation of the dopant can change the binding configuration with the environment. Such mechanisms are common for species dissolved in He droplets, leading e.g. to a migration towards the droplet surface followed by the ejection of the dopant on a pico to femtosecond time scale [64, 65].”

In addition, the title and the introduction claimed an investigation on the role of the environment, but this is not what the paper does. There is no real comparison between different environments (except that recognizing that in Ne droplets there is a contribution by inhomogeneous broadening), nor a discussion about what the environment does on the system. Cooled gas nanodroplets are not a natural environment for molecules and therefore I found it inappropriate to claim that the role of environmental parameters affecting the dynamics could be studied in these conditions. Also because in the ultracold experimental conditions applied here there is basically no change in the system dynamics (as stated on page 5). I would suggest changing the tone of the introduction because it is bringing the reader somewhere else.

As pointed out above, many important 2DES experiments focus on static properties of the sample otherwise inaccessible with other methods. According to the reviewer’s suggestion we have modified the introduction of the revised manuscript to make the achievement of our work clearer. In particular we added the statement (p. 1, para. 4): “2DES and 2D infrared spectroscopy [21] have proven as very useful in condensed phase systems to extract line shape information where other methods are not applicable or do not provide the required time-frequency resolution [22–26].”

A final comment: most of the experimental details are lacking and I do not believe that enough detail is provided in the methods for the work to be reproduced.

In order to increase the readability of the manuscript and to focus on the major results, we refrained from repeating experimental details mentioned in previous work. The individual ingredients (fluorescence-detected phase-modulated 2DES, cluster isolation spectroscopy) are described in detail in previous publications as cited in our manuscript (Refs. 6, 38, 75, 76). Furthermore, we have recently published an extended review article about our experimental apparatus to which we refer in the manuscript (Ref. 29). We made this point clearer in the revised manuscript (p.6, para. 3): “The experimental apparatus including the cluster beam preparation are described in detail in a recent review article [29].” In addition, we have substantially extended our Methods Section to include more methodologic details (see colored text passages in revised manuscript).

Minor comments:

1. I suggest specifying better in the title that nanoparticles are intended as gas nanoclusters: in material science, ‘nanoparticles’ might indicate a completely different system.

We changed the wording accordingly.

2. I found Figures 2c and 2d a bit misleading. While I understand that the scope is illustrating in general terms the meaning of the signals appearing in the 2DES maps, one would expect to see similar features also in the experimental data. Also, I found a bit confusing the labels ‘molecule 1’ and ‘molecule 2’, considering that only one molecular species was investigated here. I suggest removing these panels, not fully relevant here (since no cross-peaks were detected), and referring to previous literature (there are many clear and comprehensive reviews about the signal distribution in 2DES maps).

From the reviewer comment we conclude that Fig. 1 was meant and we have modified Fig. 1 according to the reviewer's suggestion. However, we note that cross peak information is relevant in our study: the absence of cross peaks confirms the absence of vibrational excitations as well as the absence of inter-particle couplings among co-doped species (Fig. 3). Therefore, we have not fully removed the discussion of cross peaks in Fig. 1.

3. there is a typo on page 6, second column, line 10: 'monochromoator'
The typo is fixed in the revised manuscript.

Reviewer #2 (Remarks to the Author):

Overall, this is a noteworthy and pioneering study demonstrating that local environments strongly affect the optical properties of an organic molecule chromophore. The authors use their results to report on the molecular-surface binding configurations, which is a key contribution. The authors have chosen to write their manuscript from the standpoint of spectroscopy method-development, which lowers the impact somewhat. The impact would have been higher had the authors focused their title and abstract on the scientific (new knowledge) advances with less focus on the tool used to generate that new knowledge. The method development aspect for cluster-isolation and high-resolution 2D spectrometer is very strong.

We thank the reviewer for this suggestion and have modified title, abstract and introduction in our revised manuscript accordingly.

Major comment:

- The authors compare measurements of helium nano droplets and neon nanodroplets. There are two differences in these systems. One is the obvious change of temperature. The second is a change of solvation configuration, fully encompassed or on the surface. It is not clear how the authors disentangle these two effects without a temperature-dependent study.

We agree that temperature-dependent studies are a common and frequently applied means to provide further insight into broadening mechanisms of condensed phase samples. However, for rare-gas matrices a temperature-dependent study is not possible due to the lacking access for heat injection (infrared inactivity and immediate cooling by evaporation). As such, our study resolving for the first time the phonon broadening at a given temperature represents the maximized information content achievable for isolated rare-gas nanoclusters.

We have added the following statement in our revised manuscript to make this point clearer (p. 4, para. 6): "A sample-temperature dependent study of the homogeneous linewidth would provide further insight into the phonon scattering as commonly done in the condensed phase [18]. However, the infrared inactivity and rapid evaporative cooling of the nanoclusters prevent any means for heat injection."

- The generality of the results could have been enhanced had the authors chosen to conduct measurements of at least one more molecule.

We agree that it would be interesting to characterize the chromophore-cluster interaction for other molecular and cluster species. This is on our agenda for future experiments. However, at the current state further experiments would considerably delay the publication by at least 6 more months due to a laser brake down in our lab and difficulties to get a repair during the Corona pandemic. Given these

circumstances and the timely theme of our work, we believe that it is important to share our results with the scientific community without further delay.

To account for the reviewer's comment, we have extended the comparison of our results with experiments in bulk matrices where different samples were studied (see comment #1 of reviewer #3).

Minor comments:

- page 2 the authors write: "Peaks on the diagonal position of the 2D spectra reflect the linear absorption of the system, ...". This is often repeated in many publications, but it is not strictly true for all systems. There are at least two complications: (1) The input laser pulse spectrum almost always provides different weights to the peaks because it is not fully de-convoluted. (2) ESA peaks can - in some systems - appear on or near the diagonal and thereby affect the relative peak heights. The authors could reword as, for example: "Peaks on the diagonal position of the 2D spectra mostly reflect the linear absorption..."

We thank the reviewer for noting this inaccuracy and have changed this point in the revised manuscript, e.g. (p.2 para. 4): "Peaks on the diagonal position of the 2D spectra closely resemble the linear excitation spectrum of the system".

- Fig. (1), panel (c). The caption could be enhanced by mentioning that the coupling refers to the blue dashed arrow.

According to the suggestion of reviewer #1 we have removed this item in Fig. 1 (see also minor comment #2 of reviewer #1).

- Fig. (2), panel (b). The unit of inverse centimeters is a wavenumber, not a frequency. This abuse of the words is endemic in spectroscopy, but that does not make it correct. Please adjust so that it is scientifically correct. There are at least two good options: (1) Replace the word "frequency" with "wavenumber", or (2) Adopt the approach of Tokmakoff and label as $\omega/(2\pi c)$.

We have changed this in Fig. 2,4 and S1 of the revised manuscript.

- Page 3 the authors subtly introduce a control measurement of H2Pc in 1-chloronaphthalene. At what temperature were those measurements conducted?

We have added the information to the caption of Fig. 2.

- Fig. (3). Why did the authors choose a waiting time of 1 ps? Time zero would in principle be better because the system would have not undergone any spectral diffusion at all.

We investigated the system response also at smaller waiting times which did not reveal different behavior except for the common pulse-overlap effects observed at very short waiting times (<100fs). The latter is why we have refrained from choosing the waiting time at time zero. We clarify this in the revised manuscript (p.7, para. 3): "For the measurements in Fig. 5 the population time T was scanned from 1 to 501 ps. Values T < 100 fs were omitted to avoid the influence of parasitic pulse overlap effects."

- Fig (3) panel (a). How do the authors know that the blips along the diagonal are not simply due to scatter?

To our knowledge and experimental experience scattering contributions do generally not contribute in action-detected 2DES especially if extremely dilute samples are investigated as in our case. All satellite peaks coincide with the peaks observed in light-induced fluorescence spectroscopy where

they are identified as the molecular complexes. We clarify this point in the revised manuscript(p. 3, para. 4):" whose spectral positions coincide exactly with the weak features observed in the 2D spectra. Possible contributions from sample scattering can be excluded in action-detected 2DES [54]"

- On page 4 the authors write "...except for an outlier at 15040 cm⁻¹, which is due to a nonrecurring experimental artifact." What is the meaning of this? If it only occurred once, the authors should fit and present a different dataset that does not have the artifact.

The presented data set exhibits the highest resolution and was therefore chosen. The artifact does not occur in the 1D coherence scan (Fig. 2b). We agree that our statement is confusing and we have made this point clearer in the revised manuscript(p. 4, para. 5): "This outlier is not observed in our other data (Fig. 2, 5) and is attributed to an experimental artifact."

Reviewer #3 (Remarks to the Author):

Bangert et al. report photon echo study of free base phthalocyanine in helium and neon clusters. The authors found unexpectedly long electronic coherence of hundreds of picoseconds in helium and few picoseconds in neon. The photon echo technique enables attaining the homogeneous and heterogeneous linewidth. In both environments the homogeneous linewidths were found to be about 0.5 cm⁻¹, which is very narrow. I feel these results may deserve publication in NC. However, the manuscript should be revised according to the following points.

1. The authors should remove uncorroborated claims and proposals.

In abstract and in the text the authors claim unprecedented spectral resolution without giving appropriate comparisons. I am rather sure that a factor of at least ten higher resolution was achieved in early hole burning experiments in solid matrices. Similar, the resolution in a conventional cw LIF spectrum of phthalocyanine in helium shown in Fig.1 of SI is higher than obtained in the reviewed manuscript.

We agree that higher resolution was achieved in other spectroscopy methods. However, these methods are either not applicable to the studied system (photochemical hole burning, see reviewer #1, comment #2) or do not yield the crucial information about the homogeneous linewidth (LIF measurements). To avoid confusion, we have omitted the term "unprecedented" in the revised manuscript wherever confusion may have been produced. The text passages are now (p.1, para. 5): "Our results provide insight into the molecule-surface binding configurations with a resolution far beyond the accuracy and resolution of current density functional theory approaches." And (p.6, para. 2) "The unique combination of 2DES with cluster-isolation techniques uncovered the homogeneous lineprofile beyond the ensemble average, providing insight into the molecule-surface binding configurations with very high resolution."

Throughout the manuscript the authors implied that the photon echo technique could be applied to study: ..." local configurations in photo-chemical reactions and elucidating molecular dynamics in tailored nano-environments or at functional interfaces" (Introduction)

or

" the interaction of isolated molecules with metallic and semiconducting nanoclusters, of high relevance in catalysis and biomedical imaging" (Conclusions)

or

"to explore and control many-body mechanisms of high interest in photovoltaics, such as molecular

exciton migration and annihilation processes, as well as in quantum information science which is, so far, predominately limited to atomic samples." (Conclusions)

There is not any discussion of the applicability of the photon echo technique to any of the above systems. Therefore, the above propositions sound speculative to the present referee.

I suggest the authors focus on the studied system and provide more physical insights into the observed effects

To account for the reviewer's comment, we have modified the introduction and conclusions section of the manuscript and have extended our discussion about prospective applications of our approach. In particular, we added the following statements (p.6, para. 2): "Other methods for homogeneous linewidth retrieval cannot provide the combined time-frequency resolution of 2DES. The presented high-resolution 2DES method, thus, provides a promising perspective for the analysis of molecular dynamics as a function of the local environment with femtosecond time resolution." And "The potential of studying molecular networks on nanoconfined cluster surfaces was recently demonstrated [16, 17] and the advantage of 2DES in uncovering intra-particle interactions and cooperative dynamics was proven in previous studies [72-74]."

Furthermore, we extended the discussion of the observed effects (p. 5, para. 2): "In contrast, experiments in bulk Ne matrices showed an almost one order of magnitude broader inhomogeneous linewidth of $\approx 200 \text{ cm}^{-1}$ [58]. The same behavior is observed for 3, 4, 9, 10-perylenetetracarboxylic dianhydride (PTCDA) [61]. The much broader inhomogeneous linewidth in the bulk environment may be explained by the formation of a solvation shell when the molecule is fully embedded in the bulk matrix compared to when bound to the surface of a cluster. At the same time, the homogeneous broadening of H2Pc-NeN seems in reasonable agreement with the values obtained in bulk matrices of heavier rare-gases where a factor of ≈ 5 smaller homogeneous broadening is observed [62]. The lower matrix temperature (4 K) in these experiments may explain the reduced phonon broadening than observed in our cluster samples with temperatures of $\approx 10 \text{ K}$."

Further discussions can be found in comment #5 below.

2. The principle of the two-dimensional electronic spectroscopy should be introduced in the main text that should be accessible for non-specialists.

We agree that our work addresses an interdisciplinary readership which may not be familiar with the 2DES technique. However, in order to increase the readability of the manuscript we have spared experimental details described in previous work (see also response to reviewer #1, comment #4).

To account for the reviewer's comment, we have simplified the experimental description in section II A of the main text and have added references introducing the 2DES technique. In particular, we added the following paragraph (p. 2, para. 2): "So far, 2DES is mainly performed in the condensed phase and the desired nonlinear signals are separated from the background by non-collinear four-wave mixing geometries (coherent-detected 2DES) [37]. Conversely, for the dilute cluster beam samples a collinear beam geometry is needed combined with the detection of an action signal (action-detected 2DES) [29]. In the latter, the sample is excited with a sequence of four femtosecond laser pulses (Fig. 1b) and the fourth-order light-matter response is deduced from the detected fluorescence. The pulse delays τ, t are interferometrically scanned to track the free polarization decay of the sample induced by pulse 1 and 3, respectively. Accordingly, a Fourier transform with respect to τ, t yields 2D frequency-spectra, which directly correlate the excitation (x-axis) and detection (y-axis) frequencies, while the time delay T determines the time evolution of the correlation spectra (Fig. 1c,d) [20]." And (p. 2 para. 3): "The phase modulation leads to characteristic beat notes in the

fluorescence yield based on which the linear and nonlinear signal contributions are efficiently separated and amplified using lock-in detection."

3. "A Fourier transform of the 1D coherence scans yield the linear absorption spectra of both systems (Fig. 2b). (p.3)"

I feel this is not just FT, because the frequency of the oscillations in a) is much smaller than the peak frequency in b). The authors should better explain the application of the FT.

The responsible effect for this behavior is the rotating frame detection of the signal in the time domain. We have added the following explanation to the revised manuscript to make this point clearer (Fig. 2 caption): "In return, the frequency axis in the Fourier spectra in (b) are up-shifted by the same amount to recover the absolute transition frequencies of the system". Furthermore, we have added more information in the Methods section (see comment #6 below).

4. " Its far extending horizontal and vertical tails reflect the Lorentzian lineshape and are augmented by the saturated color scale." (p.3)

I do not see any horizontal or vertical tails in Fig. 3 (a).

We meant the horizontal/vertical lines intersecting at the main peak of the 2D spectrum. A horizontal/vertical cut through the 2D spectrum reveals that these are the tails of the line shape of the main spectral peak. We modified our explanation in the manuscript to make this clearer (p. 3, para. 4): "The horizontal and vertical lines intersecting with the ZPL show the tails of the ZPL line profile, which are augmented by the saturated color scale."

5. " We conclude that the molecules are mainly immobile on the (Ne) cluster surface on the time scale of 500 ps, which was previously only indirectly confirmed for ground state molecules [57]." (p.5)

It would be nice to have some perspective on the limits imposed by the clusters and also to have measurements, such as in Fig. 5 with helium clusters. For example, although the molecules in neon are fixed in the cluster frame, the clusters nearly certainly are rotating in the laboratory frame. In case of helium clusters the molecules may rotate and translate inside owing to its superfluid nature, on the other hand the traditional phonon induced dephasing may be muted due to finite size effects.

Some more detailed discussion of the physical origin of the long coherence in clusters as compared to other previously studied condensed system will make this manuscript much stronger.

We have added the suggested discussion in the revised manuscript (p.5, para. 5): "We note, that the rotation of the cluster in the laboratory frame adds in principle another contribution to the inhomogeneous broadening due to the cluster size distribution in the probed ensemble. This broadening is however expected to be negligible due to the large moment of inertia of the clusters.

On the contrary, dopant species are highly mobile on the surface of superfluid He droplets as reflected in the formation of alkali complexes on the droplet surface [66]. Likewise, species fully immersed in the quantum fluid clusters exhibit high mobility. Their rotational motion is well studied and served as a probe of the internal temperature of He clusters [35]. The solvation mechanisms of organic chromophores in superfluid He droplets have been previously studied with high-resolution frequency-domain spectroscopy [67]. It was rationalized that solutes are embedded in a non-superfluid He solvation layer. The different configuration of the solvation complexes lead to spectral finestructures [19, 68] while the dispersive solute-cluster interaction causes asymmetric lineshapes

due to the statistical cluster size distribution [52]. The homogeneous linewidth for the ZPL of solutes in superfluid He has not been determined to date. High-resolution 2DES may solve this problem and provide new insight into the solvation mechanisms in superfluid media. For Pc-HeN an inhomogeneous linewidth of $\geq 0.1 \text{ cm}^{-1}$ was found and an estimate for the homogeneous broadening of 0.02 cm^{-1} was indirectly deduced [52]. Both are beyond our current experimental resolution. This could be solved by deploying advanced sampling strategies [69–71] or implementing frequency-comb technology [55] in order to increase the resolution to the required regime, thus, opening a promising perspective for the study of solvation mechanisms in superfluid environments.”

6. In methods, the modulation frequencies $\Omega(i)$ should be given. The explanation of the signal processing in the last two paragraphs is not understandable for a not specialist. It would be nice to have some equations for more clarity.

We have added the frequency values in the Methods section. Moreover, we extended the explanation of the signal processing and included some equations to make it clearer to the reader. We changed the following paragraph to make this clearer (p. 7, para. 5): “As reference signal for the lock-in detection, a portion of the pulse train is spectrally filtered in a monochromator (0.025 nm spectral resolution) at a frequency of $\nu_R = c/662.33 \text{ nm}$ and detected with a PMT. The reference signal records the optical interference of the pulses and, thus, tracks the phase changes and phase jitter in the optical setup at the frequency ν_R . The heterodyne demodulation of the fluorescence signal $S(\tau, T, t, \Omega \pm \Omega_3)$ with reference signal $R(\tau, T, t, \Omega \pm \Omega_3)$ inside the lock-in amplifier removes the modulation as well as the phase jitter from the signal S , which provides the required interferometric stability for the measurements. In addition, S is detected in the rotating frame of R , which down-shifts the signal frequencies ν_S by 2 orders of magnitude to $\bar{\nu} = \nu_S - \nu_R$. To recover the absolute frequencies in the Fourier spectra, the frequency axis are up-shifted again by ν_R .”

REVIEWERS' COMMENTS

Reviewer #1 (Remarks to the Author):

The authors did a major rewriting job, changing and clarifying the introduction, data description, and discussion.

I found the revised manuscript significantly improved with respect to the previous version. Many points, previously perceived as weaknesses, have been better described and clarified, and the manuscript indeed reads more solid now.

I am still not entirely convinced about the maturity of the results, and I do not yet fully agree with some of the answers to my concerns. Nonetheless, this is the risk with publications reporting potentially high-impact novel results.

Overall, I think what is presented here could contribute significantly at least to stimulate new discussions about system-bath interactions. Therefore, I will not object to the publication of the paper in Nature Comm.

Reviewer #2 (Remarks to the Author):

I have reviewed the updated manuscript and the response letter provided by the authors to the reviewer queries. The novelty of the results and focus of the manuscript are more clear than the initial submission. The authors responses and changes to the manuscript were therefore beneficial, and the manuscript appears ready for publication provided the authors fix a few remaining typographical errors.

Reviewer #3 (Remarks to the Author):

The authors have revised the manuscript according to my suggestions and have answered my questions. The manuscript is now more balanced and accessible for non-specialist.

I recommend publication in NC.

Response to reviewer comments

Manuscript NCOMMS-21-48836

Title: High-resolution two-dimensional electronic spectroscopy reveals homogeneous line profiles in isolated nanoparticles

Authors: Ulrich Bangert, Frank Stienkemeier and Lukas Bruder.

Reviewer #1 (Remarks to the Author):

The authors did a major rewriting job, changing and clarifying the introduction, data description, and discussion.

I found the revised manuscript significantly improved with respect to the previous version. Many points, previously perceived as weaknesses, have been better described and clarified, and the manuscript indeed reads more solid now.

I am still not entirely convinced about the maturity of the results, and I do not yet fully agree with some of the answers to my concerns. Nonetheless, this is the risk with publications reporting potentially high-impact novel results.

Overall, I think what is presented here could contribute significantly at least to stimulate new discussions about system-bath interactions. Therefore, I will not object to the publication of the paper in Nature Comm.

We thank the reviewer for the positive evaluation of our revised manuscript.

Reviewer #2 (Remarks to the Author):

I have reviewed the updated manuscript and the response letter provided by the authors to the reviewer queries. The novelty of the results and focus of the manuscript are more clear than the initial submission. The authors responses and changes to the manuscript were therefore beneficial, and the manuscript appears ready for publication provided the authors fix a few remaining typographical errors.

We thank the reviewer for the positive evaluation of our revised manuscript and we have corrected the typographical errors in the manuscript.

Reviewer #3 (Remarks to the Author):

The authors have revised the manuscript according to my suggestions and have answered my questions. The manuscript is now more balanced and accessible for non-specialist.

I recommend publication in NC.

We thank the reviewer for the positive evaluation of our revised manuscript.